# Regional Topological Aberrances of White Matter- and Gray Matter-Based Functional Networks for Attention Processing May Foster Traumatic Brain Injury-Related Attention Deficits in Adults

**DOI:** 10.3390/brainsci12010016

**Published:** 2021-12-24

**Authors:** Ziyan Wu, Meng Cao, Xin Di, Kai Wu, Yu Gao, Xiaobo Li

**Affiliations:** 1Department of Electrical and Computer Engineering, New Jersey Institute of Technology, Newark, NJ 07102, USA; zw255@njit.edu; 2Department of Biomedical Engineering, New Jersey Institute of Technology, Newark, NJ 07102, USA; meng.cao@njit.edu (M.C.); xin.di@njit.edu (X.D.); 3School of Biomedical Sciences and Engineering, South China University of Technology, Guangzhou 510630, China; kaiwu@scut.edu.cn; 4Department of Psychology, Brooklyn College, The City University of New York, New York, NY 11210, USA; yugao@brooklyn.cuny.edu; 5The Graduate Center, The City University of New York, New York, NY 10016, USA

**Keywords:** traumatic brain injury (TBI), post-TBI attention deficits, visual sustained-attention task (VSAT), graph theoretic technique (GTT), white matter tracts

## Abstract

Traumatic brain injury (TBI) is highly prevalent in adults. TBI-related functional brain alterations have been linked with common post-TBI neurobehavioral sequelae, with unknown neural substrates. This study examined the systems-level functional brain alterations in white matter (WM) and gray matter (GM) for visual sustained-attention processing, and their interactions and contributions to post-TBI attention deficits. Task-based functional MRI data were collected from 42 adults with TBI and 43 group-matched normal controls (NCs), and analyzed using the graph theoretic technique. Global and nodal topological properties were calculated and compared between the two groups. Correlation analyses were conducted between the neuroimaging measures that showed significant between-group differences and the behavioral symptom measures in attention domain in the groups of TBI and NCs, respectively. Significantly altered nodal efficiencies and/or degrees in several WM and GM nodes were reported in the TBI group, including the posterior corona radiata (PCR), posterior thalamic radiation (PTR), postcentral gyrus (PoG), and superior temporal sulcus (STS). Subjects with TBI also demonstrated abnormal systems-level functional synchronization between the PTR and STS in the right hemisphere, hypo-interaction between the PCR and PoG in the left hemisphere, as well as the involvement of systems-level functional aberrances in the PCR in TBI-related behavioral impairments in the attention domain. The findings of the current study suggest that TBI-related systems-level functional alterations associated with these two major-association WM tracts, and their anatomically connected GM regions may play critical role in TBI-related behavioral deficits in attention domains.

## 1. Introduction

Traumatic brain injury (TBI) is one of the major public health concerns that occurs primarily in young adults [1]. TBI-related functional brain alterations, such as disruptive functional connectivity (FC) among distributed neural networks that support cognitive processes and behavioral control, have been frequently reported and suggested to likely link with severe post-TBI neurobehavioral sequelae [2,3,4,5]. Among the severe post-TBI neurobehavioral consequences, attention deficits have been found to be most commonly reported and can persist and create lifelong challenges to the affected individuals [6,7,8]. However, brain mechanisms associated with post-TBI attention deficits have not yet been sufficiently investigated.

Although TBI-induced tissue damages in both gray matter (GM) and white matter (WM) have been widely reported, WM tracts are suggested to be particularly vulnerable to mechanical shearing and the stretch forces of TBI [9,10]. Diffusion-MRI (dMRI) studies in adults with chronical TBI have reported microstructural anomalies in various WM regions and their association with post-TBI attention deficits. Specifically, Raj et al. reported increased mean diffusivity (MD) and radial diffusivity (RD) values in the genu of corpus callosum (CC) in adult patients with chronic TBI, and their associations with altered behavioral performance in an attention task [11]. Similarly, Jun et al. demonstrated that TBI subjects with chronic post-concussive syndrome had significantly reduced fractional anisotropy (FA) in the genu of CC, relative to healthy controls, and the poorer CC alignment was significantly correlated with longer reaction times in response to the Attention Network Test (ANT) [12]. Relative to matched controls, adult patients with TBI have also been found to have significantly decreased FA in the posterior corona radiata (PCR), which strongly contributes to lower neuropsychological test scores in the attention domain [13] and poorer attentional control performance quantified using the ANT [14]. In addition, adult patients with TBI have also shown significantly lower MD in the superior longitudinal fasciculus, anterior midbody of CC, and cerebral isthmus, which were all suggested to link to post-TBI attention problems [15].

The vast majority of functional neuroimaging studies in TBI have focused on the GM regions, and only a handful of these existing GM-based functional brain studies have attempted to address the neural substrates of TBI-related attention deficits. Specifically, an early task-based functional MRI (fMRI) study reported decreased brain activation in the anterior cingulate cortex in a small sample of adult patients with moderate-to-severe TBI patients, as compared to healthy controls, during a block-designed modified stroop task [16]. Another early fMRI study reported significantly reduced neural activations in the posterior parietal cortex, frontal eye fields, and ventrolateral prefrontal cortex during attentional disengagement in adults with mild TBI, relative to group-matched controls [17]. In addition, increased fMRI activation in bilateral middle-frontal and supplementary motor cortices during visual sustained-attention processing was reported in adults with moderate to severe TBI [18]. By utilizing the functional near-infrared spectroscopy (fNIRS) technique, Hibino et al. reported significantly increased medial frontal activation and decreased lateral frontal activation in a small sample of young adult patients with severe TBI, as compared to healthy controls, during the performance of an attention task [19]. During the magnetoencephalography (MEG) recording, Petley et al. showed reduced global activation and delayed reaction times when performing a visual attention task in patients with mild TBI compared to controls [20].

By utilizing the FC technique in fMRI, Bonnelle et al. reported significant associations between altered within-default mode network FC (specifically, interactions between precuneus and other brain regions including the ventromedial prefrontal cortices, inferior parietal cortices, middle temporal and frontal gyri, thalami, and parahippocampal gyri in the bilateral hemispheres) and behavioral impairments when performing a choice reaction-time task in adult subjects with TBI [21]. Significantly increased resting-state fMRI FC within the sensorimotor network and its association with impairments in the attention domain have also been reported in a relatively larger sample of adults with moderate and severe TBI, relative to controls [22]. Similarly, an MEG study reported significantly elevated resting-state FC among frontal, parietal, and temporal areas, and their linkage with inattentive behaviors in male TBI patients, relative to a matched control group [23]. Our recent fNIRS study found that young adults with mild and moderate TBI showed significantly higher inferior frontal-occipital FC for sustained-attention processing, and the abnormally increased FC were significantly correlated with more hyperactive/impulsive symptoms in the TBI group [24].

Recently, accumulating evidence has highlighted the existence and reliability of blood oxygen level-dependent (BOLD) signal fluctuations in WM [25,26,27,28,29,30], enabling the identification of functional communications across large distances among distributed WM networks. The development of a graph theoretic technique has enabled us to assess the complex and interactive patterns of multiple remote brain regions that are affected by the diffuse axonal injury nature of TBI, which has the potential to provide informative findings in uncovering network abnormalities in TBI [31,32,33,34]. To the best of our knowledge, only one recent study has explored alterations of WM network topology in a cohort of patients with TBI, which reported enhanced FC among large-scale WM networks involving inferior fronto-occipital fasciculus, primary sensorimotor, occipital, and pre/postcentral WM networks [35]. Nevertheless, systems-level TBI-related functional brain alterations in both GM and WM, and their interactions and contributions to post-TBI attention deficits, have not yet been fully addressed.

The present study has enrolled 42 young adults with TBI and 43 group-matched controls to examine the systems-level TBI-related functional brain characteristics in both GM and WM and their interactions during visual sustained-attention processing, and their contribution to post-TBI behavioral impairments in attention domain. On the bases of findings from our and other research teams [36,37,38], we hypothesize that (a) relative to the matched controls, adults with TBI would show significant and interactive topological alterations in the WM and GM functional organizations involving attention processing, particularly in the major WM tracts that play critical role in attentional deployment (including parts of the corona radiata [14,39], thalamocortical radiation [40], and superior longitudinal fasciculus [41,42]), as well as the frontal and parietal GM regions that are subserved by these proposed WM tracts for the bottom-up and top-down attentional and cognitive control processes; and (b) the regional topological aberrances, especially those in the WM network that also significantly interact with regional GM topological anomalies, significantly contribute to elevated inattentive and/or hyperactive/impulsive behaviors in adults with TBI.

## 2. Materials and Methods

### 2.1. Participants

A total of 85 young adults (18 to 27 years of age) were involved in this study. A total of 42 (including 21 males and 21 females) had a history of TBI, and 43 (including 23 males and 20 females) were group-matched normal controls (NCs). The specific inclusion criteria for the TBI group were: having a history of one or multiple sports- or recreation-related (i.e., tobogganing/sledding, amusement attractions) TBIs clinically confirmed at least 6 months prior to the study appointment, having a non-penetrating head injury which caused diffuse brain damage (according to medical records), and having no history of diagnosis with any sub-presentation of attention-deficit/hyperactivity disorder (ADHD) prior to the first onset TBI. Specific inclusion criteria for NCs were: having no history of TBI, having no history of diagnosed ADHD (any sub-presentation), and having T-scores < 60 for inattentive, hyperactive/impulsive, and combined symptoms in Conner’s Adult ADHD Self-Reporting Rating Scales (CAARS) [43], which were administered during the study assessments. The general inclusion criteria for both subject groups were: native or fluent speakers of English, and strongly right-handed, measured using the Edinburgh Handedness Inventory [44]. None of the involved participants reported a history or current diagnosis of any neurological disorder (such as Epilepsy), severe psychiatric disorder (including Schizophrenia, Autism Spectrum Disorders, Major Depression, Anxiety, Conduct Disorder, etc.), received treatment with any stimulant or non-stimulant psychotropic medication within the month prior to testing, or having MRI constraints, such as metal implants, claustrophobia, etc.

Participants in both groups were recruited from the New Jersey Institute of Technology (NJIT) through on-campus study flyers. The demographic and clinical characteristics of the involved participants are summarized in Table 1. The study received Institutional Reviewed Board Approvals at NJIT. Written informed consents were provided by all participants.

### 2.2. Experimental Task for fMRI Acquisition

During fMRI acquisition, each subject performed a block-designed visual sustained-attention task (VSAT). The VSAT has been validated for its feasibility of measuring behavioral and functional capacity of sustained attention in both children and adults [24,45,46,47,48,49]. The detailed design of the task was described in our previous studies [24,48,49]. Briefly, it consists of five task blocks interleaved by five rest blocks (Appendix A). Each block lasts for 30 s. During the rest blocks, the participant was instructed to keep their eyes open and to remain as relaxed and motionless as possible. In each of the five task blocks, a red cross appeared in the center of the computer screen and lasted for 800 milliseconds. Then, a target sequence of three-digit sets (1-3-5, 2-5-8, 3-7-9, 5-2-7, and 6-1-8, respectively) were shown in red at the rate of one digit per 400 milliseconds. After a 1.0-s break, nine sequences of three digits, ranging from 1 to 9, appeared in black in a pseudo-random order at the rate of 400 milliseconds per digit. A 1.8-s response period ensued after each sequence. In this response period, the subject was asked to press the left button of a response box with the forefinger of their right hand when the stimulus sequence (black ones) matched the target sequence (red ones), and to press the right button with the middle finger otherwise. The total duration of the entire task was 5 min.

Prior to fMRI acquisition, a short training version of the task was provided to each participant to ensure that they understood the requirements of the task. Task performances, including response accuracy rate, omission error rate, commission error rate, and overall/correct response reaction time, were examined in each subject (as shown in Table 1).

### 2.3. Experimental Setup and MRI Data Acquisition

Before the MRI scan, a pre-metal check was completed for ensuring the safety of the experiment. Each participant was then positioned on a moveable examination table. Earplugs were offered to attenuate the scanner noise. Head motion was restrained with positioning pads. A bolster was set under the participants’ knees to help them remain still and maintain the correct position during imaging. A head coil capable of sending and receiving radio waves was placed above the participant’s head. A mirror was positioned on the head coil, allowing the participant to see the visual stimuli that were performed by screen of a computer-guided projector. A two-button response box was provided for responding to the task stimuli. In addition, a squeeze ball was provided, in case the participant wanted to alert the technologist or terminate the scan.

MRI data were collected using a 3-Tesla 32 channel Siemens TRIO (Siemens Medical System, Erlangen, Germany) scanner at the Rutgers University Brain Imaging Center. The fMRI data were obtained using a gradient echo-planar sequence with the following parameters: repetition time (TR) = 1000 ms; echo time (TE) = 28.8 ms; flip angle = 30°; field of view = 208 mm; voxel size = 1.5 × 1.5 × 2.0 mm^3^ with no gap; slice number = 55. High-resolution 3D T1-weighted structural images were collected using a magnetization-prepared rapid gradient echo sequence with the following parameters: TR = 1900 ms; TE = 2.52 ms; flip angle = 9°; field of view = 250 mm; voxel size = 1.0 × 1.0 × 1.0 mm^3^; slice number = 176.

### 2.4. Individual-Level fMRI Data Preprocessing

The fMRI data were preprocessed using the FMRIB Software Library v6.0 FEAT Toolbox (Oxford, UK, https://fsl.fmrib.ox.ac.uk/fsl/fslwiki/FSL/ accessed on 7 October 2021). An initial visual check was first applied to each set of fMRI data for any missing volumes or severe head motions. Motion artifacts were then corrected by applying the rigid-body transformations [50], with the motion parameters, including the displacement (translation along the *x*-, *y*-, and *z*-axes) and rotation around these axes, were estimated using the first volume as reference. No participant was excluded for excessive head motion, with a strict cutoff threshold of displacement = 1.0 mm. Next, the acquisition time between slices was corrected, and non-brain structures were extracted. To improve the signal-to-noise ratio, images were further smoothed with a 4-mm full-width-at-half-maximum Gaussian kernel. A high-pass temporal filter of 1/75 Hz was implemented for low-frequency noise removal. Then, each fMRI data was co-registered to the structural MRI data of the same subject and normalized into the ICBM152_T1_2mm Montreal Neurological Institute (MNI) template [50,51]. The task-based whole-brain activation map was then generated using the FMRIB’s Improved Linear Model tool [52]. Motion parameters were regressed out from each fMRI datum for residual effects removal. Finally, each Z-statistic map was cluster-thresholded with the value of Z > 2.3 and at the significance level of *p* < 0.05 [53].

### 2.5. WM Functional Network Node Selection

In order to select the nodes for the WM functional network construction, a combined power spectrum map was first generated (Figure 1A). Power spectrum, which measures the signal’s power contained in a time signal at specific oscillatory frequencies [54,55], has been identified as a unique and repeatable feature for quantifying synchronous BOLD signals in WM [56]. To generate the study cohort-specific power spectrum map, a weighted-WM mask was first created (in the MNI space) in each subject, using the segmentation tool in FreeSurfer v.6.0 (Charlestown, MA, https://surfer.nmr.mgh.harvard.edu/ accessed on 17 September 2021). Then, each weighted-WM mask was binarized using a threshold of 0.5 (a default thresholding value for mask binarization, meaning that the current voxel had a >50% possibility of being classified into WM). Next, a weighted-group-WM mask was generated by combining the binarized WM masks from all the individuals. It was then binarized using a threshold of 0.8, i.e., the current voxel had a >80% possibility of being included in the individual WM masks. Furthermore, this binarized-group-WM mask was parcellated into 48 WM tracts in the MNI space, using the Johns Hopkins University (JHU) ICBM-DTI-81 WM labels atlas [57].

For each individual, time series of the preprocessed fMRI data were extracted using the 48 parcellated WM masks, and then normalized by converting an individual’s raw score into the standard z scores to avoid outlier issues. The power spectrum value of each voxel was estimated using the fast Fourier transform under the frequency of 0.017 Hz, which has been validated by previous fMRI studies in WM [26,58]. A total of 41 WM nodes (spheres with radii = 2 mm) were then placed at the identified activation peaks (local power spectrum maximum). Considering that the myelinated axon caliber varied in different WM tracts [59], we, additionally, validated these WM nodes by overlapping each of the spherical node with its associated JHU WM tract mask. Detailed information of the 41 nodes is listed in the Appendix A.

### 2.6. WM Functional Network Construction and Topological Property Estimations

In each subject, the BOLD time series of the voxels in each of the 41 WM nodes were extracted from the preprocessed fMRI data. Then, a wavelet-based approach was applied to the signals of the 41 nodes for denoising [60]. This technique provides multi-frequency information about signals, and is known to be effective for identifying non-stationary events caused by motion and for detecting transient phenomena, such as spikes [60]. Specifically, the time series of each voxel were decomposed in the wavelet domain, using the Maximal Overlap Discrete Wavelet Transform. Wavelet scales 3, 4, and 5, which provided information on the frequency band in the 0.015–0.125 Hz range, being denoted to contain the majority of the task-related hemodynamic information [61,62,63,64]), were then reconstructed to the time series signal in each voxel and averaged within each node.

Next, a 41 × 41 FC matrix was generated for each fMRI datum using the absolute values of the Pearson’s correlation coefficients. Furthermore, an averaged FC matrix was generated among participants belonging to the groups of NC and TBI, respectively (Figure 1A), and further converted into a binary graph by using the network cost, C, as the threshold value. The network cost was defined as:(1)C=K/(N(N−1)/2),
where K is the total number of possible edges and N the total number of nodes in the network [65]. In order to identify the small-world regime [66] in both groups, we calculated two global metrics, including global-efficiency and local-efficiency, for the two groups and their node- and degree-matched regular and random networks over a wide range of the cost values from 0.1 to 0.5 using increments of 0.01. The selected cost value range was commonly suggested in previous studies, allowing for a proper estimation of the small-world properties [49,67,68]. The network global-efficiency, $E_{network-glob}\left(G\right)$, was defined as the inverse of the average characteristic path length between all nodes in the network, using the following equation [65]:(2)$$E_{network-glob}\left(G\right)=\frac{1}{n\left(n-1\right)}\sum_{i,j\in G,j\neq i}\frac{1}{d_{ij}},$$
where n is the number of network nodes, and $d_{ij}$ is the inverse of the shortest path length between nodes i and j. The network local efficiency, $E_{network-loc}$, quantifies the inverse of the shortest average path length of all neighbors of a given node among themselves, which can be calculated using the following formula [65,67,69]:(3)$$E_{network-loc}\left(G\right)=\frac{1}{n}\sum_{i\in G}E_{network-glob}\left(G_{i}\right),$$
where, $G_{i}$ represents the subnetwork that contains all neighbor nodes of node i, and $E_{network-glob}\left(G_{i}\right)$ the subnetwork global-efficiency calculated using Equation (2). A network is considered as “small-world” if it meets the criteria: $E_{network-glob}\left(G_{regular}\right)<E_{network-glob}\left(G\right)<E_{network-glob}\left(G_{random}\right)$ and $E_{network-loc}\left(G_{random}\right)<E_{network-loc}\left(G\right)<E_{network-loc}\left(G_{regular}\right)$, where $E_{network-glob}\left(G_{regular}\right)$, $E_{network-glob}\left(G_{random}\right)$, $E_{network-loc}\left(G_{regular}\right)$, and $E_{network-loc}\left(G_{random}\right)$ represent the network global-efficiency and network local-efficiency of the node- and degree-matched regular and random networks, respectively [66]. As shown in Figure 1A, we observed that the locations of the global- and local-efficiency curves of both groups were between the corresponding curves of the random and regular graphs within the range of 0.1 to 0.4.

Finally, network topological properties in each subject, including 4 global-level properties (network global-efficiency, network local-efficiency, network CC, and network degree), and 5 nodal-level properties, including nodal global- and nodal local-efficiency, nodal CC, nodal degree, and betweenness centrality (BC), were estimated and averaged over the range of the cost values from 0.1 to 0.4. Definitions regarding the network properties have been detailed in previous studies [65,67,69,70]. Briefly, the nodal-efficiency, $E_{nodal}\left(G,\;i\right)$, was a local measurement which evaluated the communication efficiency between a node i and all other nodes in the network G, by using the following equation [65]:(4)$$E_{nodal}\left(G,\;i\right)=\frac{1}{n-1}\sum_{j\in G,j\neq i}\frac{1}{d_{ij}},$$
where $d_{ij}$ was the shortest path length between nodes i and j. The nodal CC, CC(G), estimates the likelihood of whether the neighboring nodes of a node i were interconnected with each other, which was defined as [70]:(5)$$CC\left(G\right)=\frac{1}{n}\sum_{i\in G}\frac{1}{k_{i}\left(k_{i}-1\right)}\times \sum_{j,h\in G_{i}}{\left(a_{ij}a_{ih}a_{jh}\right)}^{1/3},$$
where $a_{ij}$ was the connection between nodes i and j, with the value 1 for connected and 0 for not connected, and $k_{i}$ the number of neighbors of node i. The nodal degree was defined as the number of edges connected to a node i, and the BC of a node i estimated the proportion of all the shortest paths between pairs of other nodes in the network that include that node [69].

### 2.7. GM Functional Network Node Selection

In order to select the nodes for GM functional network construction, a combined activation map was first generated based on the combination (union) of the brain clusters in the average activation maps of the groups of NC and TBI (Figure 1B). This combined activation map was then parcellated according to the FC-based Brainnetome atlas [71], which divides the whole brain GM into 210 cortical and 36 subcortical subregions. Among the 246 parcellated cortical and subcortical GM regions in the combined activation map, a total of 114 regions contained more than 100 contiguous voxels that were significantly activated during the task performance. Therefore, 114 GM nodes were placed as spheres (radius = 4 mm) at the identified activation peaks (local activation maximum). The size of the node was determined based on the estimation of the average cortical thickness of an adult human brain [72,73]. Detailed information for the 114 nodes is listed in the Appendix A.

### 2.8. GM Functional Network Construction and Topological Properties Calculation

The time series of the 114 GM nodes were first extracted from each fMRI datum. The wavelet-based approach, which has been described in Section 2.6, was applied to denoise the signals in the 114 GM nodes. Then, a 114 × 114 FC matrix for each fMRI datum was generated through averaged time series within each pair of the GM node, and the group-averaged FC matrices were constructed in the groups of NC and TBI, respectively (Figure 1B). The average FC matrix in each group was converted into a binary graph by using the network cost as the threshold value (defined in Equation (1)). The small-world regime [66] was also identified in a GM functional network analysis. As shown in Figure 1B, locations of the global- and local-efficiency curves of both groups were between the corresponding curves of the random and regular graphs within the range of 0.1 to 0.4. Thus, the GM network properties, including the 4 global-level topological properties (network global-efficiency, network local-efficiency, network CC, and network degree), and 5 nodal-level topological properties (nodal global- and nodal local-efficiency, nodal CC, nodal degree, and BC), were then estimated and averaged over the range of the cost values from 0.1 to 0.4.

### 2.9. Group-Level Analyses

#### 2.9.1. Demographic, Clinical/Behavioral, and Task-Performance Measures

The demographic, clinical/behavioral, and task-performance measures were compared between the groups of NC and TBI by using a Chi-square test for discrete variables (i.e., gender and race/ethnicity) and an independent samples *t*-test for continuous variables.

#### 2.9.2. Topological Properties of WM and GM Functional Networks

Group comparisons of the WM and GM network topological measures (including both global- and nodal-level measures) were first carried out using a one-way analysis of covariance, with gender as a fixed-effect covariate, and age, participant’s education level, and participant’s parents’ education level as random-effect covariates. For topological measures that showed significant between-group differences, post hoc t-tests were further compared between the NC and TBI groups. Multiple comparisons were controlled for the results in both steps using a Bonferroni correction [74] at α = 0.05.

#### 2.9.3. WM and GM Functional Network Interaction Analysis

For all the WM and GM nodes that reported significant between-group differences in any of the 5 nodal-level topological properties, pair-wise Pearson’s correlation analysis of each nodal-level topological measure was conducted in each diagnostic group, respectively. Multiple comparisons were controlled using Bonferroni correction [74] at α = 0.05.

#### 2.9.4. Brain–Behavior Relationship Analysis

Brain–behavior relationships were investigated in each diagnostic group using partial correlation, between the neuroimaging measures that had significant between-group differences and the raw-scores of the CAARS inattentive and hyperactive symptoms subscale scores, by controlling age, participant’s education level, and participant’s parents’ education level. Again, multiple comparisons were controlled using a Bonferroni correction [74] at α = 0.05.

## 3. Results

### 3.1. Demographic, Clinical/Behavioral, and Task-Performance Measures

Demographic characteristics and task-performance measurements showed no significant between-group differences. All participants achieved >95% responding accuracy when performing the experimental task during fMRI. Relative to NCs, individuals with TBI showed significantly higher inattentive and hyperactive symptom scores (Table 1).

### 3.2. Topological Properties of WM and GM Networks

The WM and GM network global- and local-efficiency measures did not significantly differ between the two groups. Group comparisons of the WM network nodal properties showed that, relative to NCs, individuals with TBI had a significantly higher nodal local-efficiency (*t* = 2.143, *p* = 0.035 after Bonferroni correction) in the left PCR, and a significantly lower nodal global-efficiency (*t* = −2.072, *p* = 0.042 after Bonferroni correction) in the right posterior thalamic radiation (PTR). Group comparisons of GM network nodal properties showed that, relative to controls, the TBI group had a significantly higher nodal degree (*t* = 2.246, *p* = 0.027 after Bonferroni correction) and nodal global-efficiency (*t* = 2.426, *p* = 0.036 after Bonferroni correction) in the left postcentral gyrus (PoG), and a significantly higher nodal local-efficiency (*t* = 2.016, *p* = 0.047 after Bonferroni correction) in the right superior temporal sulcus (STS).

### 3.3. Interactions of WM vs. GM Network Topological Properties

In the group of controls, nodal local-efficiency of the left PCR in WM was significantly positively correlated with the nodal global-efficiency (r = 0.330, *p* = 0.031 after Bonferroni correction) of the left PoG in GM, whereas, in the group of TBI, the nodal local-efficiency of the left PCR in WM showed a trend of significantly negative correlation with the nodal global-efficiency of the left PoG (r = −0.277, *p* = 0.075 after Bonferroni correction) in GM; the nodal global-efficiency of the right PTR in WM was significantly positively correlated with the nodal local-efficiency of the right STS (r = 0.353, *p* = 0.022 after Bonferroni correction) in GM (Figure 2).

### 3.4. Brain–Behavior Relationships

The BC of the left PCR in WM demonstrated a trend of significant negative correlation with the raw-scores of the CAARS hyperactive symptoms (r = −0.292, *p* = 0.075) in the group of TBI, however, such a pattern was not found in controls (Figure 3). No significant (or trend of significant) correlations were found in the TBI or NC groups between the CAARS inattentive symptom raw-score and the neuroimaging measures.

## 4. Discussion

The current study, for the first time in the field, investigated the functional network organizations in both WM and GM during sustained-attention processing and their interactions with post-TBI behavioral attention deficits in adults with TBI. Relative to the matched controls, individuals with TBI demonstrated an abnormally higher ability of information propagation (represented by a significantly increased nodal local-efficiency [66]) of the left PCR, as well as a lower functional integration (represented by significantly decreased nodal global-efficiency [75]) of the right PTR in WM. The TBI group also showed significantly higher functional integration and connectivity strength (represented by a significantly increased nodal global/nodal-efficiency and degree) of the left PoG and the right STS in GM, when compared with the group of controls. In the systems-level, subjects with TBI also demonstrated an abnormally strong functional synchronization between the PTR and STS in the right hemisphere, as well as a hypo-interaction between the PCR and PoG in the left hemisphere.

As the most prominent projection fiber, the PCR contains both ascending and descending fibers that connect subcortical nuclei and the primary sensory cortex, including the PoG in the parietal lobe [76,77]. The PTR, also referred to as thalamocortical radiations, connects fibers extending from subcortical regions to visual and sensorimotor cortices [78]. However, there is no evidence showing any major branches of the PTR anatomically connecting the STS. The PCR, PTR, and the cortical and subcortical GM regions anatomically connected with these two major-association WM tracts have been widely validated to play critical role in normal attentional deployment [79,80,81,82,83,84,85,86,87,88,89]. Previous dMRI studies in adults with TBI have frequently reported microstructural abnormalities, including reduced FA and increased MD and RD in the PCR and PTR [13,90,91]. Structural MRI studies have also reported regional cortical abnormalities in the PoG and STS [92,93,94]. Together with the functional aberrances (e.g., between regional or within-network FC) of these brain regions or in GM areas associated with the WM fibers that were reported in existing literatures [23,95,96], our findings suggest that TBI-related abnormal topological properties associated with the PCR and PTR in WM, the PoG and STS in GM, and their abnormal functional interactions in the WM/GM functional networks for attentional information processing, may be partially underlined by the structural anomalies in these critical WM and GM regions subserving attention and higher order cognitive information processing.

Our analyses of brain–behavior relationships showed a trend of significant contribution of decreased BC of the left PCR to elevated hyperactive/impulsive symptoms in subjects with TBI. TBI-related functional and structural alterations in cortical and subcortical regions, which are structurally connected by the PCR, and their significant involvement in altered attentional control, have been demonstrated in existing neuroimaging studies. For instance, enhanced brain activations during executive control processing were observed in multiple subregions within the frontal lobe, a key component of the ascending fibers of the PCR [76,77], in male TBI patients [84]. GM tissue integrity reduction of thalamic nuclei, which are involved in the PCR descending pathways [76,77], were found to be associated with poorer performance for attention processing in adult TBI patients [86]. Taken together, our results suggest that a reduced ability for functional information flow control (which can be measured using the BC property) in the PCR and its anatomically connected GM regions may significantly contribute to post-TBI attention problems, including behavioral hyperactivity, in adults with TBI.

There are some issues associated with this study that need to be further discussed. First, our study sample included both male and female subjects. Clinical studies have reported sex-specific patterns of post-TBI cognitive and behavioral impairments [97,98,99,100]. However, findings from neuroimaging studies on sex-related brain mechanisms associated with TBI are far from converging, with some reported differentiated values of functional or structural neuroimaging measures between males and females with TBI [101,102], while others observed no significant between-sex differences [103]. Our supplementary analyses in the clinical, behavioral, and topological measures in both WM and GM networks did not report any significant between-sex differences in the TBI or control groups. Second, among the 42 subjects in the TBI group, 24 had one TBI, and 18 had multiple TBIs. Clinical studies have examined the impact of repetitive TBI on neuropsychological and behavioral impairments and demonstrated contradictory results [104,105,106,107,108,109]. As another supplementary test, we conducted Pearson’s correlation between the behavioral scores of the CAARS inattentive and hyperactive/impulsive symptoms and the number of TBIs in the patient group and did not find any significant results. In addition, the locations of brain injury varied in our TBI subjects. All the subjects in our TBI group have been recruited from the NJIT sports teams. Most of them received a rapid forward or backward force and several had the violent blow occur to the left or right side of the brain during sports-related or recreative activities. There exists a concern in the field that different cognitive and/or behavioral problems may be caused by injuries to specific locations of the head (https://msktc.org/tbi/factsheets/Understanding-TBI/Brain-Injury-Impact-On-Individuals-Functioning, accessed on 7 October 2021). Although there is no evidence yet to suggest the direct impact of injury location on cognitive and behavioral impairments, its potential influence should be investigated in the future with a much larger study sample. Third, our study utilized FreeSurfer for brain segmentation. Though it has been identified as a powerful and robust tool for whole-brain automated segmentation, possible distortion or failure of surface segmentation and registration caused by brain lesions may still occur [110]. To guarantee a proper surface tessellation, we thus conducted an additional visual inspection on each participant’s segmentation result. Future work can concentrate on validating and comparing segmentation efficiency by using multiple pipelines (e.g., SPM12-CAT, MAPER).

## 5. Conclusions

In summary, the current study reported systems-level functional aberrances associated with the PCR and PTR in WM, as well as the PoG and STS in GM during sustained-attention processing in young adults with TBI. The functional alteration in the PCR was linked to elevated behavioral hyperactivity/impulsivity in with the group of TBI. The findings of this study may provide new insights into the understanding of neurophysiological mechanisms associated with post-TBI attention deficits.

## Figures and Tables

**Figure 1 brainsci-12-00016-f001:**
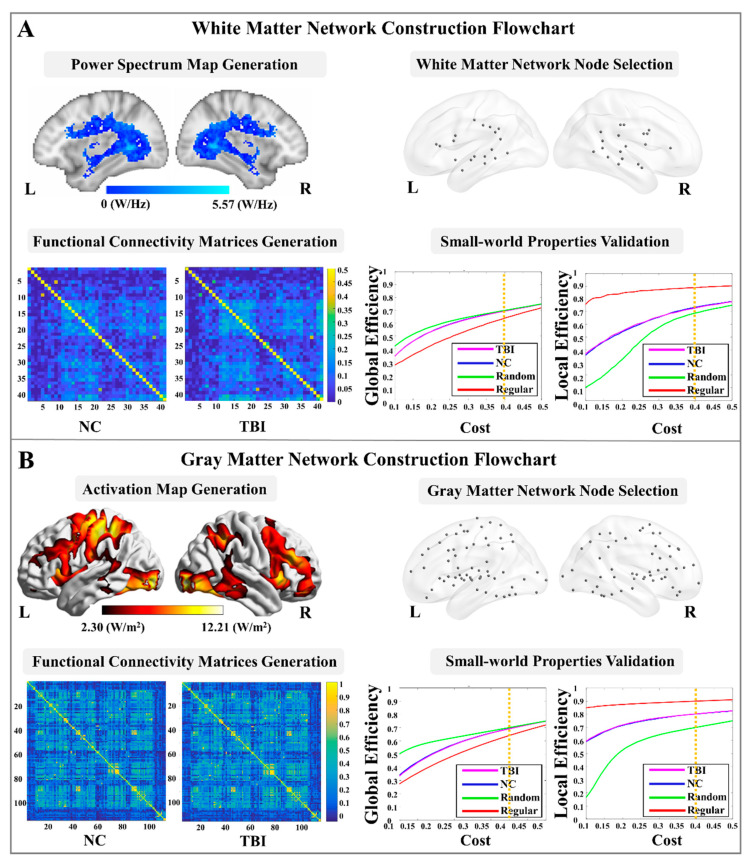
White matter and gray matter functional networks construction. (**A**) White matter network construction flowchart, including power spectrum map generation, node selection, functional connectivity matrices generation, and brain network small-world properties validation; (**B**) Gray matter network construction flowchart, including brain activation map generation, node selection, functional connectivity matrices generation, and brain network small-world properties validation. (L: left hemisphere; R: right hemisphere; NC: normal control; TBI: traumatic brain injury).

**Figure 2 brainsci-12-00016-f002:**
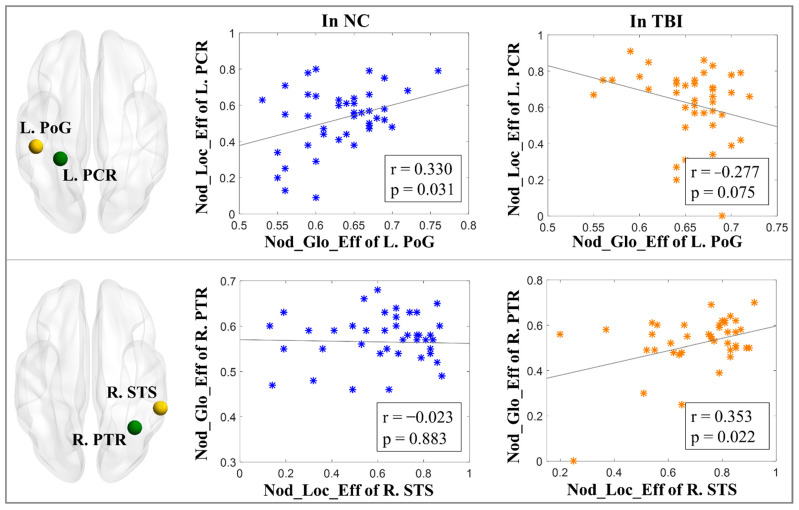
Interactions of white matter (marked in green) vs. gray matter (marked in yellow) network topological properties in the groups of NC and TBI. (NC: normal control; TBI: traumatic brain injury; r: correlation coefficient; *p*: level of significance; L.: left hemisphere; R.: right hemisphere; PoG: postcentral gyrus; PCR: posterior corona radiata; STS: superior temporal sulcus; PTR: posterior thalamic radiation; Nod_Loc_Eff: nodal local-efficiency; Nod_Glo_Eff: nodal global-efficiency.)

**Figure 3 brainsci-12-00016-f003:**
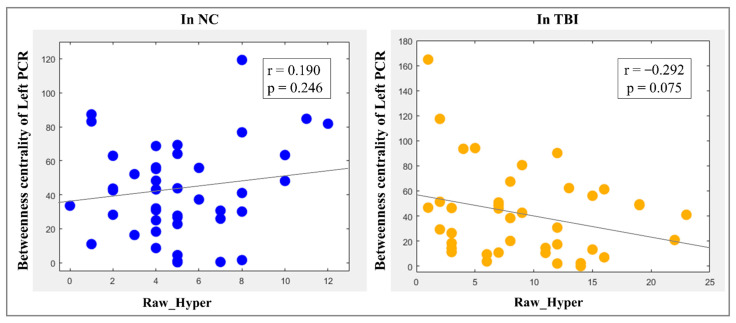
Brain–behavior relationships in the groups of NC and TBI. (NC: normal controls; TBI: traumatic brain injury; PCR: posterior corona radiata; Raw_Hyper: raw-scores of the Conner’s Adult ADHD Self-Reporting Rating Scales (CAARS) hyperactive symptoms; r: correlation coefficient; *p*: level of significance).

**Table 1 brainsci-12-00016-t001:** Demographic, clinical/behavioral, and task-performance measures of the study sample.

	NC (N = 43)	TBI (N = 42)	
	**Mean (SD)**	**Mean (SD)**	***p* Value**
Age	22.36 (2.74)	21.63 (2.00)	0.167
Education year	14.98 (1.95)	14.26 (1.56)	0.066
Mother’s education year	15.35 (2.20)	15.55 (2.70)	0.710
Father’s education year	15.77 (2.81)	15.50 (2.78)	0.660
CAARS scores			
Inattentive raw scores	4.67 (2.81)	9.31 (6.28)	<0.001
Inattentive T-scores	45.88 (6.48)	57.02 (15.18)	<0.001
Hyperactive/impulsive raw scores	5.07 (2.76)	9.19 (5.80)	<0.001
Hyperactive/impulsive T-scores	42.58 (5.93)	52.52 (14.66)	<0.001
	**N (%)**	**N (%)**	***p* Value**
Male	23 (53.49)	21 (50.00)	0.917
Right-handed	43 (100)	42 (100)	1.000
Race/Ethnicity			0.094
Caucasian	12 (27.91)	21 (50.00)	
Black or African American	4 (9.30)	7 (2.38)	
Asian	20 (46.51)	9 (21.43)	
Hispanic/Latino	2 (4.65)	2 (4.76)	
More than one race	5 (11.63)	3 (7.14)	
Functional MRI task performance measures	**Mean (SD)**	**Mean (SD)**	***p* Value**
Accuracy rate	0.99 (0.04)	0.99 (0.01)	0.344
Omission error rate	0.009 (0.03)	0.001 (0.005)	0.131
Commission error rate	0.003 (0.01)	0.005 (0.01)	0.464
Overall response reaction time (ms)	607.09 (134.73)	604.45 (132.83)	0.928
Correct response reaction time(ms)	606.72 (135.02)	603.83 (132.47)	0.921

NC: normal control; TBI: traumatic brain injury; N: number of subjects; SD: standard deviation; *p*: level of significance; CAARS: Conner’s Adult ADHD Self-Reporting Rating Scales; ms: milliseconds.

## Data Availability

Data are available upon reasonable request to the corresponding author.

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
