# Peer review of "Regional Topological Aberrances of White Matter- and Gray Matter-Based Functional Networks for Attention Processing May Foster Traumatic Brain Injury-Related Attention Deficits in Adults"

_brainsci, 2021, doi:10.3390/brainsci12010016_

Round 1

Reviewer 1 Report

This study examines the TBI-related topological alterations in the grey and white matter and their interactions during visual sustained attention processing, and their contribution to post-TBI behaviour impairments related to elevated inattentiveness and hyperactivity/impulsive behaviours. 

The key questions that I have for the authors pertain to the brain-behaviour relationships that were investigated:

1) Were any individuals screened for the NC group excluded from the study due to not meeting the inclusion criteria? In particular, it would be of interest if any candidates for the NC were excluded because of a high CAARS score, given that this score was used to evaluate the brain-behaviour relationship.

2) The methods section 2.9.4 describes that brain-behaviour relationships were investigated for a correlation between neuroimaging results and the raw scores of the CAARS for both inattentive and hyperactivity subscales. However, in the results, only the correlation with the hyperactivity symptoms is presented.

3) The reporting of this brain-behaviour relationship describes that there is a "trend of significant negative correlation", with p=0.075. It is important that the implications of this finding is not overstated, given that statistical significance has not been reached. With that in mind, the wording in the discussion that "decreased BC of left PCR was correlated with elevated hyperactive/impulsive symptoms" and that "this is the first time in the field demonstrating the linkage..." may overemphasise the results and may be misleading to the reader.

Author Response

Manuscript ID: brainsci-1512180: Regional Topological Aberrances of White Matter- and Gray Matter-based Functional Networks for Attention Processing may Foster Traumatic Brain

Injury-Related Attention Deficits in Adults

We appreciate the very detailed and constructive comments from the reviewer. We have addressed all the comments and have made specific changes in the revised manuscript accordingly. Our responses to the reviewer’s comments are below in Bold:

Reviewer 1

This study examines the TBI-related topological alterations in the grey and white matter and their interactions during visual sustained attention processing, and their contribution to post-TBI behaviour impairments related to elevated inattentiveness and hyperactivity/impulsive behaviours.

The key questions that I have for the authors pertain to the brain-behaviour relationships that were investigated:

  1. Were any individuals screened for the NC group excluded from the study due to not meeting the inclusion criteria? In particular, it would be of interest if any candidates for the NC were excluded because of a high CAARS score, given that this score was used to evaluate the brain-behaviour relationship.

During recruitment, there were no participants eligible in the NC group were excluded due to a high CAARS score.  Here we provide the plots of the CAARS Inattentive and Hyperactive/Impulsive symptom T-scores of all the 43 participants in the NC group for additional reference (Please see the attached word document).

  1. The methods section 2.9.4 describes that brain-behaviour relationships were investigated for a correlation between neuroimaging results and the raw scores of the CAARS for both inattentive and hyperactivity subscales. However, in the results, only the correlation with the hyperactivity symptoms is presented.

We are grateful for this valuable comment and apologize for the insufficient descriptions in our original manuscript. We did carry out the correlation analysis between both inattentive and hyperactive symptoms scores and neuroimaging measures. Results didn’t show any significant or trends of significant correlations between the inattentive symptom score and neuroimaging measures in both group. We have added the descriptions in Section 3.4. (page 10): “No significant (or trend of significant) correlations were found in the TBI or NC groups between the CAARS inattentive symptom raw-score and neuroimaging measures.

  1. The reporting of this brain-behaviour relationship describes that there is a "trend of significant negative correlation", with p=0.075. It is important that the implications of this finding is not overstated, given that statistical significance has not been reached. With that in mind, the wording in the discussion that "decreased BC of left PCR was correlated with elevated hyperactive/impulsive symptoms" and that "this is the first time in the field demonstrating the linkage..." may overemphasize the results and may be misleading to the reader.

We appreciate this valuable comment and fully agree with the reviewer. We have revised the manuscript in Section 4 (page 12): “Our analyses of brain-behavior relationships showed a trend of significant contribution of decreased BC of left PCR to elevated hyperactive/impulsive symptoms in subjects with TBI.

Reviewer 2 Report

The present manuscript includes a study about traumatic brain injuries using functional MRI and functional connectivity. The work is well organized and well presented. However, there are some issues that the authors need to take of great care.

Introduction, paragraph no3, the authors need to add studies that deal with such a brain insult but use different modalities and employ functional connectivity. 

Methods, section 2.4, 2nd paragraph, 

In this place, the authors have to be aware of segmentation errors that might be revealed by freesurfer and add some sentences in the discussion talking about the future possibility of using other pipelines eg SPM12-CAT, and so on.

Later on, in the same paragraph, It is not justified well why the authors selected this threshold (a>50). Is there any statistical or other scientific reason behind it?

Figure 1: Units in legends are not clear, for example, Power Spectrum Map Generation, also the resolution of the image is not sufficient for reading 

Section 2.9.3, Person's correlation a standard method however issues like heteroscedasticity are not checked. It would be better to consider such a test by employing robust statistics. If we check the later on the figure (Fig. 2) something like this is observed for the TBI group.

Discussion, paragraph no2, 

A comparison with other modalities (eg MEG with which authors use FC for the same purposes are missing). 

Author Response

Manuscript ID: brainsci-1512180: Regional Topological Aberrances of White Matter- and Gray Matter-based Functional Networks for Attention Processing may Foster Traumatic Brain Injury-Related Attention Deficits in Adults

We sincerely thank the reviewer for the valuable comments. We have addressed all the comments and have made specific changes in the revised manuscript accordingly. Our responses to the reviewer’s comments are below in Bold:

Reviewer: 2

The present manuscript includes a study about traumatic brain injuries using functional MRI and functional connectivity. The work is well organized and well presented. However, there are some issues that the authors need to take of great care.

  1. Introduction, paragraph no3, the authors need to add studies that deal with such a brain insult but use different modalities and employ functional connectivity.

We greatly appreciate this valuable suggestion and have revised the Introduction section accordingly (page 2). A few other relevant neuroimaging studies are also included: “… During the magnetoencephalography (MEG) recording, Petley et al. showed reduced global activation and delayed reaction times when performing a visual attention task in patients with mild TBI compared to controls [20]…”  and “…Similarly, an MEG study reported significantly elevated resting-state FC among frontal, parietal and temporal areas and their linkage with inattentive behaviors in male TBI patients, relative to matched control group [23]…

References

20        Petley, L.; Bardouille, T.; Chiasson, D.; Froese, P.; Patterson, S.; Newman, A.; Omisade, A.; Beyea, S. Attentional dysfunction and recovery in concussion: effects on the P300m and contingent magnetic variation. Brain Inj 2018, 32, 464-473, doi:10.1080/02699052.2018.1429022.

23        Dunkley, B.T.; Da Costa, L.; Bethune, A.; Jetly, R.; Pang, E.W.; Taylor, M.J.; Doesburg, S.M. Low-frequency connectivity is associated with mild traumatic brain injury. Neuroimage Clin 2015, 7, 611-621, doi:10.1016/j.nicl.2015.02.020.

  1. Methods, section 2.4, 2nd paragraph,

2.1 In this place, the authors have to be aware of segmentation errors that might be revealed by freesurfer and add some sentences in the discussion talking about the future possibility of using other pipelines eg SPM12-CAT, and so on.

We are grateful for this valuable comment and have added this potential concern in the revised manuscript in Discussion (page 12): “Third, our study utilized FreeSurfer for brain segmentation. Though it has been identified as a powerful and robust tool for whole-brain automated segmentation, possible distortion or failure of surface segmentation and registration caused by brain lesions may still occur [110]. To guarantee a proper surface tessellation, we thus conducted additional visual inspection on each participant’s segmentation result. Future work can concentrate on validating and comparing segmentation efficiency by using multiple pipelines (e.g. SPM12-CAT, MAPER).”.

Reference

  1. Siegel, J.S.; Shulman, G.L.; Corbetta, M. Measuring functional connectivity in stroke: Approaches and considerations. J Cereb Blood Flow Metab 2017, 37, 2665-2678, doi:10.1177/0271678X17709198.

2.2. Later on, in the same paragraph, It is not justified well why the authors selected this threshold (a>50). Is there any statistical or other scientific reason behind it?

We sincerely appreciate the reviewer’s comment and apologize for the insufficient descriptions in our original manuscript. Yes, it is a default thresholding value for mask binarization. Supportive materials can be found through this weblink (https://surfer.nmr.mgh.harvard.edu/fswiki/mri_binarize). We have added the corresponding contents in Materials and Methods Section 2.5. (page 6): “Then each weighted-WM mask was binarized using a threshold of 0.5 (a default thresholding value for mask binarization meaning that the current voxel had a >50% possibility for being classified into WM)”.

  1. Figure 1: Units in legends are not clear, for example, Power Spectrum Map Generation, also the resolution of the image is not sufficient for reading

We have added the units of power spectrum and light intensity in the Figure 1 and improved the resolution of the figure.

  1. Section 2.9.3, Person's correlation a standard method however issues like heteroscedasticity are not checked. It would be better to consider such a test by employing robust statistics. If we check the later on the figure (Fig. 2) something like this is observed for the TBI group.

We are grateful for this valuable comment. Following the suggestion, we carried out the Levene test between the involved GM and WM measures in each of the TBI and NC groups. We found that the distributions of the nodal degree of left PoG and the nodal local efficiency of left PCR in the TBI group were significantly different (p=0.01). All the other measures involved in the correlation analyses were homogeneously distributed. The related text and Figure 2 are now revised accordingly.

  1. Discussion, paragraph no2, A comparison with other modalities (eg MEG with which authors use FC for the same purposes are missing).

Following this suggestion, we have revised the second paragraph of the Discussion section accordingly (page 11): “Together with the functional aberrances (e.g. between regional or within network FC) of these brain regions or in GM areas associated with the WM fibers that were reported in existing literatures [23,95,96], our findings suggest that”.

References

  1. Dunkley, B.T.; Da Costa, L.; Bethune, A.; Jetly, R.; Pang, E.W.; Taylor, M.J.; Doesburg, S.M. Low-frequency connectivity is associated with mild traumatic brain injury. Neuroimage Clin 2015, 7, 611-621, doi:10.1016/j.nicl.2015.02.020.

  1. Antonakakis, M.; Dimitriadis, S.I.; Zervakis, M.; Papanicolaou, A.C.; Zouridakis, G. Altered Rich-Club and Frequen-cy-Dependent Subnetwork Organization in Mild Traumatic Brain Injury: A MEG Resting-State Study. Front Hum Neurosci 2017, 11, 416, doi:10.3389/fnhum.2017.00416.

  1. Antonakakis, M.; Dimitriadis, S.I.; Zervakis, M.; Papanicolaou, A.C.; Zouridakis, G. Aberrant Whole-Brain Transitions and Dynamics of Spontaneous Network Microstates in Mild Traumatic Brain Injury. Front Comput Neurosci 2019, 13, 90, doi:10.3389/fncom.2019.00090.
